# Energetic Compounds in the Trophic Chain—A Pilot Study Examining the Exposure Risk of Common Eiders (*Somateria mollissima*) to TNT, Its Metabolites, and By-Products

**DOI:** 10.3390/toxics10110685

**Published:** 2022-11-12

**Authors:** Luca Aroha Schick, Jennifer Susanne Strehse, Tobias Hartwig Bünning, Edmund Maser, Ursula Siebert

**Affiliations:** 1Institute for Terrestrial and Aquatic Wildlife Research, University of Veterinary Medicine Hannover, Foundation, Werftstraße 6, 25761 Büsum, Germany; 2Institute of Toxicology and Pharmacology for Natural Scientists, University Medical School Schleswig-Holstein, Campus Kiel, Brunswiker Straße 10, 24105 Kiel, Germany

**Keywords:** dumped munitions, TNT toxicity, food chain, Common Eider, marine environment

## Abstract

The Baltic and North Seas still contain large amounts of dumped munitions from both World Wars. The exposure of the munition shells to the seawater causes corrosion, which leads to the disintegration of shells and a leakage of energetic compounds, including the highly toxic 2,4,6-trinitrotoluene (TNT), and consequently threatening the marine environment. To evaluate the risk of accumulation of energetic compounds from conventional munitions in the marine food chain, we analyzed the presence of TNT and its metabolites 2-amino-4,6-dinitrotoluene (2-ADNT) and 4-amino-2,6-dinitrotoluene (4-ADNT) as well as their byproducts 1,3-dinitrobenzene (1,3-DNB) and 2,4-dinitrotoluene (2,4-DNT) in different tissues (including muscle, liver, kidney, brain, and bile) from 25 Common Eiders (*Somateria mollissima*) from the Danish Baltic Sea. Tissues were prepared according to approved protocols, followed by GC-MS/MS analysis. None of the aforementioned energetic compounds were detected in any of the samples. This pilot study is one of the first analyzing the presence of explosive chemicals in tissues from a free-ranging predatory species. This study highlights the need for continuous monitoring at different levels of the trophic chain to increase our knowledge on the distribution and possible accumulation of energetic compounds in the marine environment in order to provide reliable data for decision-making tools and risk assessments.

## 1. Introduction

The issue of dumped munitions remaining on the seafloor of the North Sea and Baltic Sea has become an increasing concern to environmental researchers, policy-makers, and the general public during the last few years [1,2,3]. According to estimates, 300.000 tonnes of conventional munitions are still present in the German waters of the Baltic Sea alone, and a further 1.3 million tonnes are in the North Sea, not taking into account an additional 200.000 tonnes (or more) of chemical munitions along the German coast [1]. Other major dumping sites in the Baltic Sea include the Bornholm basin in the east, off Poland and Sweden, and the Skagerrak straight in the North adjacent to Norway and Denmark, pointing out the international relevance to surrounding countries [4].

The development of monitoring and detection methods is of high relevance to enhancing the localization and mapping of munitions on the seafloor, especially since their exact location remains largely unknown. Threats originating from unexploded ordinances include accidental encounters, e.g., with stranded munitions or in fishing gear [5], and the leakage and distribution of chemicals into the environment [6,7,8,9]. The latter is enhanced by continuous corrosion of the shells and poses a threat to the marine environment and biota, as well as seafood consumers [3,10]. The corrosion rate of dumped munitions is influenced by different environmental factors like salinity, water temperature, UV radiation, and the water currents [11,12]. Therefore, climate change may have an indirect impact on corrosion [12] and cause even greater concern in the near future. 

The detection of 2,4,6-trinitrotoluene (TNT) and its derivates 2-amino-4,6,-dinitrotoluene (2-ADNT) and 4-amino-2,6-dinitrotoluene (4-ADNT) in water, sediments [13,14], and marine biota [8,15] close to munition dumping grounds proves the release of energetic compounds to the environment. Subsequent studies investigating the risk of bioaccumulation and transfer along the trophic chain, as well as the threat to seafood consumers, are still lacking [10]. Initial studies on TNT in aquatic organisms indicate that the potential for accumulation and transfer to other organisms via food intake is low due to rapid biotransformation; however, no information about the levels in apex predators is available, and data on wild animals and samples obtained directly from the marine environment are also lacking [16,17]. 

Due to the high toxicity potential of TNT and its derivatives, a better understanding of their distribution in the environment is essential. TNT is genotoxic in fish [18], microorganisms, and mammals [19], and can impair larval development in Marine Mussels (*Mytilus galloprovincialis*) [20]. Furthermore, hepatotoxicity, immunotoxicity, and anemia commonly occurred in experimental studies in different mammals, such as rodents and dogs [21,22,23,24]. While very few studies have reported the toxicity of TNT in birds, the toxic effects observed are similar to reports from other organisms; lethal effects were observed at high doses, whereas impaired liver and kidney function and changes in hematological parameters were observed at sublethal high doses [25,26].

The sheer amount of munitions and the accompanying risks for the environment inevitably raise the issue of how to deal with the remains, which Koch et al. [11] address. The authors described different scenarios, from a “repository” to a “full clean up”. However, the removal is regarded as complicated and dangerous to the operators due to continuous corrosion. “Blast-in-place” operations do not solve the pollution problem as pieces of the energetic materials still contain toxic components, and blasting may even lead to their sudden or increased release into the environment [27]. Furthermore, the pressure and sound waves from detonation may cause harm to the marine fauna, e.g., causing hearing loss or blast injuries in Harbor Porpoises [28]. 

This study is one of the first approaches to detect energetic compounds from dumped munitions in a predatory species from the Baltic Sea [29] and the first to use GC-MS/MS analysis. Different tissues from 25 Common Eiders (*Somateria mollissima*) were analyzed for the presence of TNT, its derivates 2-ADNT and 4-ADNT and their byproducts 1,3-DNB and 2,4-DNT to estimate their potential accumulation along the trophic chain. In order to address the diverse problems and the different solutions for handling dumped munitions, it is necessary to fully understand the extent of the distribution of energetic compounds in the environment and at the different trophic levels. As the Common Eider mainly feeds on Blue Mussels (*Mytilus edulis*) [30], and the presence of energetic compounds has already been reported for these mollusks [8,27], an investigation on the possible accumulation of TNT and its metabolites along the trophic chain is appropriate. Furthermore, Common Eiders are still commonly hunted in northern European countries [31], thus the results from this study could provide further information for risk analyses, not only for marine fauna but also for human seafood and game consumers, and can contribute to action plans for the future management of dumped munitions.

## 2. Materials and Methods

Organ material was collected from 25 Common Eiders, which were dissected and examined as part of a pathological study [32]. All animals originate from the Sound (Øresund), a strait that connects the Baltic Sea with the Kattegat (Figure 1) and were captured incidentally (bycaught) in gillnet fishing gear between the 24 January 2017 and 28 February 2019. All captures of Common Eiders were incidental, i.e., the birds drowned as a result of the fishing activity in the area targeting other animals (fish in this case). The collection of carcasses was done opportunistically and none of the animals were euthanized or killed for the study. The animals were collected within 24 h after the nets were hauled out, then stored at −20 °C and thawed for approximately 36 h at room temperature prior to dissection. Samples of liver, kidneys, muscle, subcutaneous fat, and brain were collected in plastic bags and frozen at −20 °C. Bile samples were collected from six animals and frozen at −20 °C in urine monovettes. Sex and age were determined during the necropsy of the Common Eiders, and the nutritional condition was categorized (mortally emaciated, poor, moderate, and good) [33]. 

Table 1 gives an overview of the Common Eiders, basic information, and the conducted analyses for each animal. The samples were analyzed by two different approaches: For the first method, 1.0 g of tissue (liver, kidney, muscle, fat, and brain) was placed in a beaker and freeze-dried for approximately 24 hrs in an Alpha 2–4 LSCplus freeze dryer (Martin Christ Gefriertrocknungsanlagen GmbH, Osterode, Germany). The samples were grinded, and 0.1 g tissue of each animal was placed in a 50 mL polypropylene tube. After adding 1 mL acetonitrile and 10 µL of a stable isotope-labelled internal standard (^13^C^15^N-TNT and ^13^C-1,3-DNB, 100 ng/mL each in acetonitrile, Cambridge Isotope Laboratories, Inc., Tewksbury, MA, USA), tubes were mixed for one minute using a VF2 vortex mixer (Ika Works Inc., Staufen im Breisgau, Germany), sonicated for 30 min in a Bandelin Sonorex Super RK 510 H (BANDELIN electronic GmbH and Co. KG, Berlin, Germany), and mixed again for another minute. Afterwards, the samples were centrifuged for 10 min at 4100 rpm (4 °C) in a Heraeus Megafuge 11 R centrifuge (Thermo Fisher Scientific Inc., Waltham, MA, USA). Supernatants were decanted and diluted with 10 mL ultrapure water (18.2 MΩ cm, ELGA PURELAB flex, Veolia Water Technologies GmbH, Celle, Germany). Solid-phase extraction was carried out using Chromabond Easy solid-phase extraction columns (3 mL, 200 mg, MACHEREY-NAGEL GmbH & Co. KG, Düren, Germany). To elute the constituents, the columns were flushed with 3 mL of acetonitrile. The extract was concentrated to 1 mL using a Christ RVC 2-25 CDplus rotary vacuum concentrator (Martin Christ Gefriertrocknungsanlagen GmbH, Osterode, Germany) and transferred to 1.5 mL autosampler vials. The recovery rate of TNT and its metabolites ranges between 65 and 85%.

For the second approach, 100 mg of frozen liver or kidney tissue was transferred to a 2.0 mL Eppendorf tube and 3600 units’ β-Glucuronidase (*Helix pomatia* Type H-1, 2,274,000 units/g, Sigma-Aldrich Chemie GmbH, Taufkirchen, Germany), 400 µL sodium acetate buffer, and 10 µL of stable isotope labelled internal standards were added. β-Glucuronidase was added to biochemically cleave any glucuronidation conjugate of the energetic compounds that may have occurred during phase II metabolism in the Common Eiders. The tubes were incubated at 37 °C and mixed at 300 rpm in an Eppendorf Thermomixer compact (Eppendorf SE, Hamburg, Germany) over night and centrifuged for 0.5 h at 14,800 rpm (4 °C) in a Heraeus Fresco 21 Centrifuge (Thermo Fisher Scientific Inc., Waltham, MA, USA).

To the supernatant, 1.0 mL ultrapure water was added. From this, 1.0 mL was pipetted onto a Chromabond Easy column under a mild vacuum for solid phase extraction (3 mL, 200 mg, MACHEREY-NAGEL GmbH and Co. KG, Düren, Germany). This step was repeated five times or until the supernatant was clear. The remaining supernatant was then also pipetted onto the column. The columns were first rinsed with 3.0 mL ultrapure water and flushed with 3.0 mL acetonitrile to elute the constituents. The extracts were concentrated by rotary vacuum concentration to 1 mL and transferred to 1.5 mL autosampler vials.

The bile samples were processed accordingly using 100 µL of bile and 3600 units of β-Glucuronidase (*Helix pomatia* Type H-1, 2,274,000 units/g, Sigma-Aldrich Chemie GmbH, Taufkirchen, Germany) and in a second approach 3600 units β-Glucuronidase (*Patella vulgata* Type L-II, 2,149,000 units/g, Sigma-Aldrich Chemie GmbH, Taufkirchen, Germany).

The GC-MS/MS analyses were performed as described in Bünning et al. [35]: A Thermo Scientific TRACE 1310 gas chromatograph, coupled to a TSQ 8000 EVO triple quadrupole mass spectrometer with an electron ionization source, was used. The GC was equipped with a TraceGold TG-5MS amine 15 m × 0.25 mm × 0.25 µm column (Thermo Fisher Scientific Inc., Waltham, MA, USA). Injections were performed automatically into a programmable temperature vaporization (PTV) injector with packed quartz wool liners (2 mm × 2.75 mm × 120 mm, Thermo Fisher Scientific Inc., Waltham, MA, USA) by a TriPlus 100 LS autosampler: Helium served as carrier gas for the GC with a flow rate of 1.2 mL × min^−1^, and argon as collision gas for the mass spectrometer (both Alphagaz, purity 99.999%). A volume of 5 µL of sample was injected for each measurement. The following oven temperature programs were used: 100 °C (1 min), 35 °C/min to 220 °C (0.7 min), 70 °C to 280 °C (1 min). The total run time of each measurement was 6.99 min. Spectra were recorded in multiple reaction monitoring modes and analyzed with Chromeleon 7.2 (Thermo Fisher Scientific Inc., Waltham, MA, USA). The limits of detection (LoD) and limits of quantification (LoQ) are derived from the method-specific ones described in Bünning et al. 2021 and listed in Table 2.

## 3. Results and Discussion

In this study, we analyzed tissue samples from 25 Common Eiders for the presence of toxic substances from conventional munitions of the First and Second World Wars.

Ten animals were female and 15 were male. The majority of the female animals were adults (*n* = 7), two were subadults, and one was juvenile. The same was true for the males, with nine adults, two subadults, three juveniles, and one immature. Ten animals were in good nutritional status, nine were moderate, five were poor, and one was very poor (Table 1). 

All of the examined substances, namely 2,4,6-trinitrotoluene (TNT) and its metabolites 2-amino-4,6-dinitrotoluene (2-ADNT) and 4-amino-2,6-dinitrotoluene (4-ADNT), as well as two byproducts, 1,3-dinitrobenzene (1,3-DNB) and 2,4-dinitrotoluene (2,4-DNT) were below the limit of detection in the analyzed tissue and bile samples of all Common Eiders. 

The problem of marine-dumped munitions and energetic compounds leaking into the environment from corroding munitions has received increased attention over the past decades. Nowadays, it is common knowledge that energetic compounds have started to dissolve into water, sediment, and marine biota [3,6]. An ongoing improvement of detection methods [35] has made it possible to measure even very low amounts of TNT and related substances. To date, the aforementioned compounds have been detected, i.e., in water, sediment, mussels, and fish [6,8,15,36], leading to a growing concern about environmental impact as well as health and safety for seafood consumers [10]. In a preliminary study at the Kolberger Heide, algae, tunicates, and star-fish were manually sampled by divers within close proximity of discarded mines and from a BIP crater site within 1 m of submerged munitions [14]. Concentrations of munitions compounds in the different species were reported to range between 1 and 100 μg/kg and up to 24 μg/g in single species [14]. Finally, flatfish (Dab; *Limanda limanda*) were caught, and their bile was analyzed for the presence of munitions compounds in order to improve knowledge about the degree and spatial distribution of fish contamination originating from that dumpsite [15]. Concentrations of explosives in the bile were rather low, ranging from zero up to the low ng/per mL range [15].

This is one of the first studies analyzing tissues of marine top predators for compounds of dumped munitions and thus investigating their presence in the trophic chain. A study by Slobodnik et al. [29] screened tissues of marine mammals from the Baltic (Harbor Porpoise (*Phocoena phocoena*), Common Dolphin (*Delphinus delphis*), Harbor Seal (*Phoca vitulina*) and Grey Seal (*Halichoerus grypus*)) for different substances from dumped munitions, including the ones investigated in this study. They could not detect any of the substances, however, their LoD was much higher, with 2 ng/g wet weight (ww) for TNT and 15 ng/g ww for 2- and 4-ADNT, respectively, while the LoD in our study was 0.5 ng/g dry weight (dw) for TNT and 0.1 ng/g dw for 2- and 4-ADNT. 

When analyzing environmental pollutants in top marine predators, the high mobility of the animals is always a limiting factor. Common Eiders migrate over long distances, and although their home range is quite defined during the wintering or breeding season [37], recent feeding activity cannot be confined to a certain location. The Baltic/Wadden Sea flyway population inhabits an area from the Dutch Wadden Sea to the northern Baltic coast [38,39,40]. Male adults, especially, travel large distances and may cover migration routes from the Netherlands up to Finland [39,41,42]. Main wintering grounds on these routes, among others, can be found in the German Baltic Sea along the federal state of Schleswig-Holstein [42,43], an area which also holds some major munition dumping sites [1,34,44]. 

Common Eiders mainly feed on blue mussels, which makes them suitable for analyses and risk assessment, as concentrations of energetic compounds in blue mussels from the Baltic Sea exist [6,8,27]. In the Kolberger Heide, a major study site for sea-dumped munition in the south-west of the Baltic Sea, biomonitoring was performed with caged blue mussels [8,9,27]. In short, moorings with two mussel bags, one at the ground and one at a height of one meter, filled with 20 mussels each, were placed at a depth of approximately 11 m in close proximity either to corroding moored mines or, alternatively, to blast craters with explosive material (hexanite) scattered in the vicinity as a result of selective blasting. The exposure period was around three months. As a result, in the mussels deployed at the moored mines, 4-ADNT was detected at up to 8 µg/kg ww [9]. The concentrations of 4-ADNT did not differ significantly regarding the distance to the mine mount or regarding the position of the mussels on the mooring. Interestingly, for mussels deployed at the sediment surface near scattered chunks of munitions, the concentrations of TNT, 2-ADNT, and 4-ADNT were 31.04, 103.75, and 131.31 µg/kg, respectively [27]. 

Common Eiders are benthic feeders and prefer depths of 10 to 20 m for foraging, although deep dives up to 43 m are possible and have been recorded as well [45,46]. In the central Baltic and Atlantic the water depth of munition dumpsites varies between 50 and several hundred meters, but on-route dumping grounds in shallow waters and major munition dumpsites close to shore measure depths below 20 m [6,11]. This makes them accessible for Common Eiders and puts them at risk of taking up food in the direct vicinity of munition dumpsites. 

The estimated daily food intake of Common Eiders is 2000 g, which equals 660 g of mussel meat when solely feeding on mussels [47]. Based on a concentration of 31 ng/g ww TNT in Blue Mussels in direct vicinity of a bulk of explosive [8] and an average bird weight of 2.8 kg (personal observation) this results in a maximum daily intake of 0.007 mg/kg/d. 

Exposure studies on energetic compounds in birds are rare, but a study by Gogal et al. [25] calculated a NOAEL of 7 mg TNT/kg/d based on the observed effects in the Northern Bobwhite (*Colinus virginianus*). Lethal effects have been observed in Northern Bobwhite and Common Pigeons (*Columba livia*) at acute doses of 2.003 mg/kg and daily oral doses of 200 mg/kg/d respectively [25,26]. Despite the limited data situation, we calculated a margin of exposure (MoE) of 1000, based on the NOAEL of Northern Bobwhites, to give an estimate of the risk for Common Eiders. According to the European Food Safety Authority (EFSA) a MoE of 10,000 or higher is of low concern [48]. It has to be noted that the calculations rely on a few measurements of TNT and present a worst-case scenario. Furthermore, the NOAEL is based on a single study, and inter-species differences could not be taken into account due to a lack of data. Nonetheless, this low MoE points out that the risk for Common Eiders should not be underestimated and monitoring of marine predatory species should be continued.

An apparent limitation of this work is the sampling site with regard to the distance from known munition dumping areas. Greinert et al. (2019) [6] have recently provided a regional-scale distribution of dissolved TNT in bottom waters throughout the southwest of the German Baltic Sea and have shown clear gradients in TNT concentration ranging between 0 pM and 354 pM. Areas with munitions dumpsites (e.g., Kolberger Heide; Lübecker Bight) or known munitions contamination tended to show the highest concentration, but with a nearly order-of-magnitude difference between maximum concentrations observed in the Kolberger Heide (354 pM) compared with Lübecker Bight (43 pM). TNT concentrations were lowest in the Arkona Basin and the Mecklenburger Bight, likely as a result of their deeper water columns and greater distances from munitions sites. Water exchange through the central channel and Belt Sea (north and northeast of Fehmarn) further dilutes contamination originating from dumpsites. The widespread presence of munitions in the Baltic Sea water column makes it challenging to link specific TNT sources to the presence of TNT in any biota, including Common Eiders. In addition to long-range transports and dilution of TNT through currents, microbial or abiotic degradation or sorption on sedimentation particles have to be considered [49,50,51,52]. 

The detection of energetic compounds in the environment and mollusks seems to correlate with their distance to dumping sites [6,27] and concentrations rapidly decrease with growing distance. Maximum concentrations in water of 3100 µg/L declined to 3.3 µg/L only 50 cm away from the source [49]. In combination with the low water solubility of TNT (130 mg/L) [53], the compounds are quickly diluted. Furthermore, TNT seems to be eliminated rather quickly by effective biotransformation capabilities [17,54,55], and therefore has a low accumulation potential despite its lipophilic character [56,57]. These characteristics, combined with the high mobility of the target organisms, make it difficult to measure energetic compounds in top predators, and a measurable uptake seems most likely if the animals recently fed in close proximity to areas contaminated with munitions. 

However, this poses a problem when collecting samples of marine predators. In this study, the opportunistic collection of the animals from fishermen did not allow a broader distribution of sampling sites. Additionally, the Common Eider is protected in some European countries (e.g., Germany) [58], and therefore, animals cannot be hunted or collected at any designated location. 

Unfortunately, Sletten Haven is not located in direct vicinity to known munition dumping areas [11,34], therefore short-term exposure of the studied animals is quite unlikely. It cannot be ruled out that the Common Eiders fed closer to areas with submerged munitions at some point due to their migratory behavior and common wintering grounds along the Baltic Sea coastline. We could not detect the examined energetic compounds in any of the tissues of 25 Common Eiders or the bile samples of six animals, which would point towards accumulation in the trophic chain. However, we cannot rule out that the negative results are linked to low or even nonexistent pollution of their recent prey. 

Nonetheless, our analyses provide the first evidence that a widespread accumulative risk for marine top predators does not exist to date. To examine whether the distance to areas with dumped munition and the time during and after exposure play a role in the toxic burden of marine top predators, more animals from known munition dumping areas should be examined. By including dietary or toxicological analyses of recently ingested food, the primary uptake of energetic substances could be further evaluated. Analyses of stomach contents could also provide information about long-term exposure to low levels of energetic compounds, which are not measurable in the tissue of predatory species but still pose a health risk due to the repeated uptake of low doses. Furthermore, telemetry data of individuals could provide information about common feeding grounds and a general distance to munition dumping sites.

As mentioned before, experimental studies have shown rapid excretion of TNT and its metabolites in different organisms and it is generally believed that the bioaccumulation potential is low [9,17,59]. Therefore, one may argue that the risk for top predators and human seafood consumers is low, despite findings of energetic compounds at lower levels of the marine food chain. Even though we could not provide evidence for TNT exposure in the investigated samples, a calculable risk for Common Eiders as an exemplary species for marine top predators exists. 

To date, the impact of the chronic contamination of the environment and its ecosystems is poorly understood. The ongoing corrosion of submerged munitions housings will inevitably lead to more leakage of toxic substances into the environment and latent exposure of the organisms living within. It is also important to keep in mind that no action has yet been taken to effectively clean up the seabed of dumped munitions [11,44]. The past has also shown that the Baltic Sea is prone to accumulative effects of pollution due to its enclosed surroundings and few connections, which allow turnover of water masses [60,61]. Anthropogenic pollutants with high bioaccumulation potential, e.g., polychlorinated biphenyls, are detectable even years after their complete ban [61,62]. Furthermore, the fate of submerged munitions and behavior of the therefrom released chemicals in the environment may be influenced by unpredictable factors like climate change [12].

Fortunately, the issue of submerged munitions has received more attention in the past few years, and some countries are developing monitoring guidelines and risk-assessment strategies to establish ongoing surveillance of areas contaminated with munitions, the quantity of remaining munitions, their integrity, and the risks arising from them [6,44,63]. Such risk assessment is also based on data from energetic compounds measured in the environment. To improve the reliability of such systems, more data needs to be gathered from different areas and more species to provide a comprehensive picture of the presence of energetic compounds in the marine ecosystem.

Our study provides some of the first data for top marine predators, which is crucial to evaluating the risk for marine species as well as for human seafood consumers. However, more samples are needed to gather reliable results, and therefore top predators such as the Common Eider should be investigated for energetic compounds in future studies. The sampling site may not be ideal however, it is the first study analyzing energetic compounds at such low detection levels in a top predator and will act as a baseline for future studies or provide a negative control area that can be used as a comparison to other areas.

## Figures and Tables

**Figure 1 toxics-10-00685-f001:**
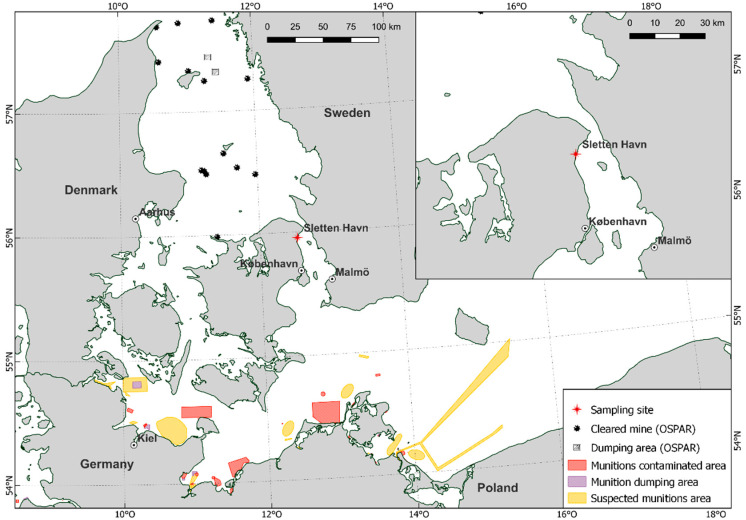
Location of Sletten Haven, where all Common Eiders were collected, as well as the presence of known munition sites in the surrounding waters of the Baltic Sea and Kattegat [34].

**Table 1 toxics-10-00685-t001:** Basic information of the examined animals and overview of the conducted analyses for each animal, sorted by collection date.

Animal Data					Sample Preparation				
Animal ID	Collection Date	Sex	Age	Nutritional State	Liver	Kidney	Muscle	Brain	Bile
Sm1	24.01.2017	male	juvenile	moderate	❄	❄	❄		
Sm2	24.01.2017	female	subadult	moderate	❄; β-PV	❄; β-PV	❄	❄	β-PV; β-HP
Sm3	05.04.2017	male	immature	moderate	❄	❄	❄		
Sm4	05.04.2017	female	adult	good	❄	❄	❄		
Sm5	05.04.2017	male	subadult	moderate	❄	❄	❄		
Sm6	05.04.2017	male	adult	good	❄	❄	❄		
Sm7	03.11.2017	female	juvenile	bad	❄	❄	❄		
Sm8	10.01.2018	male	adult	bad	❄	❄	❄		
Sm9	19.01.2018	male	adult	bad	❄; β-PV	❄; β-PV	❄	❄	β-PV; β-HP
Sm10	19.01.2018	female	adult	bad	❄	❄	❄		
Sm11	19.01.2018	female	adult	good	❄	❄	❄		
Sm12	19.01.2018	male	subadult	moderate	❄	❄	❄		
Sm13	19.01.2018	female	adult	good	❄; β-PV	❄; β-PV	❄	❄	β-PV; β-HP
Sm14	23.03.2018	female	subadult	good	❄	❄	❄		
Sm15	06.09.2018	male	juvenile	good	❄; β-PV	❄; β-PV	❄	❄	β-PV; β-HP
Sm16	27.11.2018	male	adult	good	❄; β-PV	❄; β-PV	❄	❄	β-PV; β-HP
Sm17	07.12.2018	male	adult	very bad	❄	❄	❄		
Sm18	07.12.2018	male	juvenile	bad	❄	❄	❄		
Sm19	18.01.2019	male	adult	moderate	❄	❄	❄		
Sm20	06.02.2019	male	adult	moderate	❄	❄	❄		
Sm21	13.02.2019	male	adult	moderate	❄	❄	❄		
Sm22	27.02.2019	male	adult	moderate	❄	❄	❄		
Sm23	27.02.2019	female	adult	good	❄; β-PV	❄; β-PV	❄	❄	β-PV; β-HP
Sm24	28.02.2019	female	adult	good	❄	❄	❄		
Sm25	nb	female	adult	good	❄	❄	❄		

❄ = freeze dried; β-PV = Treated with with ß-glucuronidase from *Patella vulgata*; β-HP = Treated with β-glucuronidase from *Helix pomatia.*

**Table 2 toxics-10-00685-t002:** Calculated limits of detection (LoD) and limits of quantification (LoQ) of the examined energetic compounds per g of prepared tissue sample or mL of bile, respectively, according to Bünning et al., 2021.

	Compound				
	TNT	1,3-DNB	2,4-DNT	2-ADNT	4-ADNT
**LoD (ng/g dw; ng/mL)**	0.5	0.3	0.1	0.1	0.1
**LOQ (ng/g dw; ng/mL)**	1.6	1.1	0.4	0.3	0.4

## Data Availability

All data generated or analyzed during this study are included in this published article [and its Supplementary information files].

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
