# Peer review of "Energetic Compounds in the Trophic Chain—A Pilot Study Examining the Exposure Risk of Common Eiders (Somateria mollissima) to TNT, Its Metabolites, and By-Products"

_toxics, 2022, doi:10.3390/toxics10110685_

Round 1

Reviewer 1 Report (Previous Reviewer 1)

All comments have been taken into account. Thanks to the authors for the revision and prompt correction

Author Response

Thank you very much for your final statement!

We are very happy that we could make all the required changes and adapt the manuscript according to you expectations.

Reviewer 2 Report (Previous Reviewer 2)

The text has been significantly improved. My doubts have been discussed and partly explained in the text. I believe that the publication can be accepted as preliminary or pilot research. Nevertheless, such information should already appear in the title and in the abstract.

I encourage the authors to take into account my suggestions regarding the methodology in further research, because while the preliminary research can be based on certain simplifications, the subsequent stages should already take into account environmental factors, which I discussed in more detail in the previous review.

Author Response

Thank you very much for your comment! 

We are happy that we could make all the required changes and the manuscript now meets the expectations. We have also changed the title and abstract to indicate clearly that this is a pilot study.

We will certainly keep all the suggestions in mind and hope that we can implement them in future projects.

This manuscript is a resubmission of an earlier submission. The following is a list of the peer review reports and author responses from that submission.

Round 1

Reviewer 1 Report

For the completeness of the work, data on the content of detentions in WATER are needed.

Water solubility data for TNT and others are also needed.

We need calculated data on LD50 for various objects (ideally, according to those described in the article)

If the authors did not find traces of analyzed substances in the objects, then they probably did not exist in the source water, or it is necessary to study the objects to the trophic level.

Reviewer 2 Report

The subject taken up by the authors is very interesting, as it concerns the difficult problem of potential contamination of the Baltic waters with the remains of explosives from World War II. It is no secret that flooded ammunition containers deteriorate and the risk of the release of toxic substances increases every year. It is also a well-known fact that the concentration of a number of toxic compounds increases further down the trophic chain. Therefore, I consider studies on the potential spread of these pollutants, especially in the waters of the Baltic Sea, which is an inland sea with a relatively low water exchange with the ocean, as an important trend in monitoring works.

The obtained concentration values below the limit of quantification using the technique with a very low threshold of quantification (GC-MS), with professional preparation of samples for chemical analyzes (extraction, concentration of the sample, etc.) may suggest that the problem is under control so far and even negligible. Nevertheless, a closer reading of the text raises some serious doubts as to the findings obtained. Moreover, these doubts are raised by the authors themselves in the chapter "discussion of the results".

The place and method of sampling are the biggest concerns regarding the credibility and significance of the obtained results. The eider trapping point near Sletten Havn is a minimum of 150 km as the crow flies from the nearest areas potentially at risk of contamination with munitions residues. To this should be added the fact that eiderdowns are considered as middle-distance migrants, so their food may come from many places - geographically even more distant from the discussed sources of pollution.  Moreover, the chemical compounds studied by the authors themselves are degraded (as the authors themselves write about). Scientific reports can be found about the biodegradation of TNT when exposed to environmental factors over a period of weeks. In addition, the authors themselves admit that the number of birds examined is relatively low for a monitoring study of such potential validity. There is also a lack of analyzes of water / sediments and/or tissues of low-mobility invertebrates (which are the food of the studied birds) of the discussed region.

Therefore, I consider the presented research incomplete, and even misleading, as it may suggest that the risk of pollution, if any, is negligible.

I suggest two ways to improve the quality of research. In both, it would be necessary, in my opinion, to expand the area and scope of research.

The first way, the research should be supplemented with the analysis of water, sediments and tissues of animals with much lower mobility (molluscs, crustaceans) - from the area of the current sampling. This would give an answer whether, despite such a distance (more than 150 km) from potential sources of pollution, these compounds can be found in this particular place. If so, a further conclusion could be that despite contamination, the wide area of bird nutrition causes no accumulation to be observed. It also opens the field for further studies of the dependence of concentrations on the distance.  If the concentrations of TNT and the other explosives in water/ sediments and the tissues of invertebrates were also below the limit of quantification, then it can be concluded that the mobility of the tested compounds is really low and at such a distance the concentrations are negligibly low.

The second possible scenario is to extend bird surveys to other areas of the Baltic coast to screen them as well. Certainly, the areas closer to the potential sources of pollution should be examined. Then even negative results (concentrations below the limit of quantification) would entitle to draw broader conclusions.

Some authors also include studies on concentrations of heavy metals such as Pb and Hg - because the compounds of these metals were and are often used as primary explosives (doi.org/10.3389/fmars.2018.00141) - such analyzes can also be taken into account by the authors when extending the research.

To sum up, the manuscript in the presented form should not be published in my opinion. The presented research can be considered as an introduction / some background to subsequent analyzes that should be carried out in order for the text to be suitable for publication in the Toxics journal. The ways I have proposed to extend the research are, of course, only a suggestion. Authors may choose a different path. Nevertheless, in my opinion, it is necessary to add more samples for analysis and then reformatting the text.

Reviewer 3 Report

Huge amounts of munitions have been dumped in the Baltic Sea, and it is a matter of concern that residues from these may pose a risk to the marine fauna. In this paper various organs from common eider have been analyzed for TNT and TNT metabolites to investigate potential accumulation of the substances. Since the dumped munition has started to leak it is of high importance to investigate these issues. A lot of samples have been analyzed so the effort has been substantial, and even though no TNT residues were detected it is important also to publish negative findings. One problem with the work, however, is that the birds were collected in an area quite far away from the known dumping sites (Fig 1 in the article) and their likely feeding area. The authors, therefore, need to provide better rational for the selection of animals (a sort of proof-of-concept work although nothing was proofed…), and also provide in the discussion better arguments for the relevance of the selection of animals. You give better arguments for the opposite in the text. The rational for using common eider appears, however, reasonable.

A few questions pop up in my head, which can be relevant to discuss:

-          Over how large areas do common eider feed when settled in an area?

-          The relevance to use common eider as a bioindicator for TNT exposure?

-          Other more relevant birds that are more resident?

-          How much mussels do the birds eat in the context of measured TNT levels in mussels at contaminated sites (e.g. 100 ng/g mussel) – this may indicate potential risk of effects and how likely it is to detect exposure.

-          Does common eider feed at dumping sites or is it more reasons to believe that it only will happens occasionally due to the heterogeneously spread of the dumping sites?

In the introduction I also miss some information about the dumping sites, such as on how deep waters the munitions are dumped and dumping areas. This is of relevance when selecting indicator species and their feeding area. I also miss information of actual concentrations measured in the fauna close to the dumping sites, which gives a useful context in the interpretation of the risks of exposure.

Some additional comments:

The information provided under “Institutional Review Board Statement” about the capture of the eider should also be provided under the method section.

Recovery data of the TNT and the metabolites should be included in the method.

Table 3 is not necessary; it is enough to refer non-detect in the text.

Reviewer 4 Report

The paper is very interesting from scientific point of view. The results clearly indicated that energetic compound(s) and its metabolites decomposes completety in trophic chain. The results are very important because describes situation in Baltic Sea which is relatively small and closed sea.